Excessive G–U transversions in novel allele variants in SARS-CoV-2 genomes

Panchin Alexander Y. alexpanchin@yahoo.com
Panchin Yuri V.
Institute for Information Transmission Problems, Russian Academy of Sciences , Moscow , Russia
Grande-Pérez Ana
Electronic publication date: 2020 Jul 28
Publication date: 2020
Volume: 8
Electronic Location ID: e9648
Received 2020 May 24; Accepted 2020 Jul 13
Copyright: © 2020 Panchin and Panchin
Copyright year: 2020
Copyright holder: Panchin and Panchin
License: This is an open access article distributed under the terms of the Creative Commons Attribution License, which permits unrestricted use, distribution, reproduction and adaptation in any medium and for any purpose provided that it is properly attributed. For attribution, the original author(s), title, publication source (PeerJ) and either DOI or URL of the article must be cited.
License URL: https://creativecommons.org/licenses/by/4.0/

Keywords: SARS-CoV-2, COVID-19, Mutations, Transversions, Evolution, Mutagenesis, Bioinformatics

Funding: Russian Foundation for Basic Research (RFBR) 18-29-13014 mk This work was supported by the Russian Foundation for Basic Research grant RFBR 18-29-13014 mk. The funders had no role in study design, data collection and analysis, decision to publish, or preparation of the manuscript.

==============================
Background

SARS-CoV-2 is a novel coronavirus that causes COVID-19 infection, with a closest known relative found in bats. For this virus, hundreds of genomes have been sequenced. This data provides insights into SARS-CoV-2 adaptations, determinants of pathogenicity and mutation patterns. A comparison between patterns of mutations that occurred before and after SARS-CoV-2 jumped to human hosts may reveal important evolutionary consequences of zoonotic transmission.

Methods

We used publically available complete genomes of SARS-CoV-2 to calculate relative frequencies of single nucleotide variations. These frequencies were compared with relative substitutions frequencies between SARS-CoV-2 and related animal coronaviruses. A similar analysis was performed for human coronaviruses SARS-CoV and HKU1.

Results

We found a 9-fold excess of G–U transversions among SARS-CoV-2 mutations over relative substitution frequencies between SARS-CoV-2 and a close relative coronavirus from bats (RaTG13). This suggests that mutation patterns of SARS-CoV-2 have changed after transmission to humans. The excess of G–U transversions was much smaller in a similar analysis for SARS-CoV and non-existent for HKU1. Remarkably, we did not find a similar excess of complementary C–A mutations in SARS-CoV-2. We discuss possible explanations for these observations.

Introduction

SARS-CoV-2 is a novel coronavirus that causes an infectious respiratory disease called COVID-19 (Wu et al., 2020). SARS-CoV-2 is closely related to the bat coronavirus RaTG13 with around 96% whole genome nucleotide sequence identity (Zhou et al., 2020). At the complete genome scale it also shares 93.3% identity with the bat-derived coronavirus RmYN02, with 97.2% identity in the 1ab gene (Zhou et al., 2020). Since the discovery of SARS-CoV-2, its evolution became of particular interest to biologists because of the amount of sequencing data that was produced and the importance of this data to provide insights on the determinants of viral pathogenicity (Gussow et al., 2020) and adaptations to human hosts (Van Dorp et al., 2020). SARS-CoV-2 is also quite interesting from a genomics prospective, with an extreme deficiency of genomic CpG dinucleotides of debatable origin (Xia, 2020).

Comparative analysis of genomic data favors the natural origin of SARS-CoV-2, accompanied by natural selection either before or after zoonotic transfer directly from bats or through an intermediate host (Andersen et al., 2020). In either scenario, SARS-CoV-2 would be exposed to both a novel evolutionary landscape that affects the fitness of its genetic variants and novel cellular conditions that could affect its mutation rates directly. For example, it is known, that bats have evolved a number of adaptations including superior resistance to oxidative stress (Chionh et al., 2019) that allow them to harbor multiple viruses without getting the corresponding diseases (Schountz et al., 2017). In light of this, we decided to investigate if the relative mutation frequencies in SARS-CoV-2 changed after its transmission to human hosts.

We compared the relative frequencies of single nucleotide variations (which we will refer to as mutations) in SARS-CoV-2 with the relative frequencies of substitutions that it acquired since the divergence with its last common ancestor with a closely related coronavirus from bats RaTG13. In our terminology, substitutions occurred before zoonotic transmission, while mutations were acquired after. A similar analysis was performed for SARS-CoV and HKU1 coronaviruses.

We found that SARS-CoV-2 has an over 9-fold excess of G–U mutations over substitutions. This effect was much weaker in a similar analysis for SARS-CoV and was not present for HKU1. On the other hand, the substitution profile of SARS-CoV-2 turned out to be quite similar to that of the other coronaviruses, lending further support to existing scenarios of its natural origin (Andersen et al., 2020) and suggesting that the changes in SARS-CoV-2 mutation frequencies have accompanied its transition to human hosts.

Methods

Mutation data

We obtained 1,271 SARS-CoV-2, 194 SARS-CoV and 38 HKU1 publicly available complete genomes from the NCBI NR database. To ensure similarity in data acquisition, we did this by using the BLASTn (Altschul et al., 1990) program with the three reference human coronavirus genomes as queries (NCBI Reference Sequences: NC_045512.2, NC_004718.3 and NC_006577.2). SARS-CoV-2 hits were filtered using the following strings: “Severe acute respiratory syndrome-related coronavirus” and “complete genome”. SARS-CoV hits were filtered using the strings: “SARS coronavirus” and “complete genome”. HKU1 genomes were filtered using strings “HKU1” and “complete genome”. The final list of obtained accessions is available in Table S1.

We used Clustal Omega (Sievers et al., 2011) to create three multiple alignments: one for SARS-CoV-2, one for SARS-CoV and one for HKU1 genomes. In each alignment, we established the consensus sequence. The most frequent nucleotide in each position was used as the consensus nucleotide. If any individual sequence contained a nucleotide N2 that is different from the consensus N1, we considered that a N1–>N2 mutation has occurred in that position. We assume that these mutations have occurred in viruses after zoonotic transfer. Note that if several sequences contained the same nucleotide N2 that is different from the consensus N1, this would still count as only one mutation. The three final alignments are available in fasta format in Supplemental Materials.

As additional controls, we performed the following (and only the following) subanalysis. For SARS-CoV-2: SARS-CoV-2 genomes that were sequenced with Ion Torrent

SARS-CoV-2 genomes that were sequenced with Oxford Nanopore

SARS-CoV-2 genomes that were sequenced in USA

SARS-CoV-2 genomes that were sequenced in China

Exclude mutations that occurred in less than two sequences

Substitute the consensus sequences with the reference sequence

Mask the first and last 100 nucleotides of the alignment

Remove genomes with “N” sequences

Substitution data

For each of the three human coronaviruses we obtained the genome of a closely related animal coronavirus and a more distant (outgroup) coronavirus from NCBI NR. For SARS-CoV-2 we used RaTG13 (GenBank: MN996532.1) as the close relative and pangolin coronavirus PCoV_GX-P5L (GenBank: MT040335.1) as an outgroup. For SARS-CoV we used Bat SARS-like coronavirus isolate Rs4231 (GenBank: KY417146.1) and Bat coronavirus BtCoV/273/2005 (GenBank: DQ648856.1). For HKU1 we used Betacoronavirus sp. strain VZ_BetaCoV_16715_52 (GenBank: MH687968.1) and Camel coronavirus HKU23 Ry123 (GenBank: KT368891.1). Multiple alignments were performed using Clustal Omega with default parameters (Sievers et al., 2011). See Fig. S1 for a simple Neighbor-joining tree for all nine different complete coronavirus genomes created by Clustal Omega on the basis of the alignment (default parameters). See Jaimes et al. (2020) for a detailed phylogenetic analysis of SARS-CoV-2 and related coronaviruses.

Substitutions that occurred during human coronavirus evolution were identified by maximum parsimony. If a human coronavirus sequence contained a nucleotide N1 and the two animal coronaviruses contained a different nucleotide N2, we considered that an N2–>N1 substitution has occurred (Method 1). We assume that these substitutions have occurred in viruses before zoonotic transfer. We did not count positions in which all three coronaviral genomes differed.

We also used a different measure of substitutions based on single nucleotide differences between each reference human coronavirus sequence and RaTG13, Rs4231 and Betacoronavirus sp. strain VZ_BetaCoV_16715_52 for SARS-CoV-2, SARS-CoV and HKU1 respectively (Method 2). This method does not allow us to establish a direction for substitutions, but provides a larger sample size. We used the same alignment as in Method 1.

Results and Discussion

We identified 1,251, 1,128 and 2,039 single nucleotide variants in available SARS-CoV-2, SARS-CoV and HKU1 genomes. We assumed that these are mutations that occurred after zoonotic transfer to human hosts. Mutations are deviations of individual genomes from the consensus sequences, but it should be noted that our SARS-CoV-2 consensus had only five nucleotide differences from the reference genome.

In the corresponding genomes we also identified 450, 499 and 3,029 (Method 1, with parsimony-reconstructed ancestral states) and 1,141, 1,237 and 6,514 (Method 2) single nucleotide substitutions. We assume that these substitutions occurred before zoonotic transfer. Substitutions are deviations in the reference genome from the predicted ancestral states (Method 1) or deviations in the reference genome from a closely related animal coronavirus genome (Method 2).

Figure 1 shows the proportion of each mutation/substitution type (substitutions based on Method 1) among all mutations/substitutions. The relative substitution frequencies between SARS-CoV-2 and RaTG13 appear to be similar to those between other human coronaviruses and their relatives, consistent with the natural emergence of the SARS-CoV-2 virus (Andersen et al., 2020).

Figure 1 Fraction of each mutations and substitutions in three human coronaviruses.

Mutations are deviations of individual genomes from the consensus sequences. Substitutions are deviations in the reference genome from the predicted ancestral states.

Tables 1 and 2 give a more detailed view of the number of mutations and substitutions in SARS-CoV-2 (Methods 1 and 2 respectively). P-values are based on a two-tailed Fishers exact test.

Table 1 Number and relative proportions of mutations and substitutions (Method 1) in SARS-CoV-2.

P-values are calculated with a two-tailed Fishers exact test and are not corrected for multiple comparisons.

Type	Mutations	Substitutions	% Mutations	% Substitutions	Excess of mutations	P-value	
A–C	30	5	2.4	1.1	2.16	0.121	
A–G	128	68	10.2	15.1	0.68	0.007	
A–U	40	18	3.2	4.0	0.80	0.449	
C–A	38	9	3.0	2.0	1.52	0.315	
C–G	10	4	0.8	0.9	0.90	0.770	
C–U	460	118	36.8	26.2	1.40	<0.0001	
G–A	114	29	9.1	6.4	1.41	0.092	
G–C	35	2	2.8	0.4	6.29	0.002	
G–U	193	7	15.4	1.6	9.92	<0.0001	
U–A	42	27	3.4	6.0	0.56	0.0179	
U–C	129	157	10.3	34.9	0.30	<0.0001	
U–G	32	6	2.6	1.3	1.92	0.191	

Table 2 Number and relative proportions of mutations and substitutions (Method 2) in SARS-CoV-2.

P-values are calculated with a two-tailed Fishers exact test and are not corrected for multiple comparisons.

Type	Mutations	Substitutions	% Mutations	% Substitutions	Excess of mutations	P-value	
A–C	30	18	2.4	1.6	1.52	0.189	
A–G	128	136	10.2	11.9	0.86	0.192	
A–U	40	47	3.2	4.1	0.78	0.232	
C–A	38	21	3.0	1.8	1.65	0.065	
C–G	10	10	0.8	0.9	0.91	1.000	
C–U	460	351	36.8	30.8	1.20	0.002	
G–A	114	126	9.1	11.0	0.83	0.118	
G–C	35	6	2.8	0.5	5.32	<0.0001	
G–U	193	20	15.4	1.8	8.80	<0.0001	
U–A	42	59	3.4	5.2	0.65	0.032	
U–C	129	331	10.3	29.0	0.36	<0.0001	
U–G	32	16	2.6	1.4	1.82	0.057	

Out of 1,251 SARS-CoV-2 novel variants, 193 (15.4%) are G–U transversions, which is over 9.5-fold greater comparing to 7/450 (1.56%, Method 1) or 20/1,141 (1.75%, Method 2) G–U substitutions (P < 0.0001/12, two-tailed Fishers exact test, with Bonferroni correction for 12 multiple comparisons).

This effect was not found for SARS-CoV (41/1,128 or 3.63% mutations vs. 11/499 or 2.2% substitutions, Method 1 vs. 33/1,237 or 2.67% substitutions, Method 2) or HKU1 (89/2,039 or 4.36% mutations vs. 253/3,029 or 8.35% substitutions, Method 1 vs. 625/6,514 or 9.59% substitutions, Method 2).

Most SARS-CoV-2 genomes are currently sequenced using Illumina platforms. To rule out possible sequencing artifacts we performed several subanalysis. First, we searched for mutations in SARS-CoV-2 genomes that were sequenced using Ion Torrent (30 genomes) or Oxford Nanopore (192 genomes). Although the mutation sample size was small, the fraction of G–U mutations was similar to the rest of the data: 13/79 or 16.4% (Ion Torrent) and 30/199 or 15.1% (Oxford Nanopore) G–U mutations.

Rayko & Komissarov (2020) have reported a lower transition/transversion ratio in singleton SARS-CoV-2 genomic variations (that can only be seen in one genome submission). As a separate control, we looked at mutations that were identified in two or more independent genome assemblies. This yielded a similar high proportion of G–U mutations (49/360 or 13.6%). Masking of the first and last 100 nucleotides of the alignments or removing all sequences with at least one “N” letter resulted in 16% (188/1,173) or 14.87% (132/888) G–U mutations correspondingly. De Maio et al. (2020) reported other sequencing issues of SARS-CoV-2 genomes, such as particular highly-mutable sites that might be recurring artifacts. However, the number of such reported sites is too low to affect our results as we counted several single nucleotide variations of the same type in one site as the result of one mutation.

We wanted to see if the effect is robust to other subsamples, such as SARS-CoV-2 genomes sequenced in USA or China. For USA genomes, we obtained 15.1% G–U mutations (159/1,053). For Chinese genomes, the mutation sample size was very low, however the effect was similar (20.5% or 17 out of 83 mutations are G–U).

Among the 193 G–U mutations in SARS-CoV-2, 21 are outside of coding regions, 21 are synonymous and 151 are nonsynonymous. We found no nonsense G–U mutations. Coordinates of all G–U mutations are available in Table S2.

There are two remarkable observations regarding the excess of G–U transversions in SARS-CoV-2. One is that it probably reflects a change in SARS-CoV-2 mutation rates after zoonotic transfer to humans, since the proportion of G–U substitutions measured between the SARS-CoV-2 and the bat coronavirus RaTG13 is unremarkable.

The second remarkable feature is that this excess of mutations is asymmetric: there is no similar effect for C–A mutations. SARS-CoV-2 is a positive (+) RNA strand virus. The copying of positive and negative strands of coronavirus RNA is executed by the same enzymes (Sola et al., 2015). If RNA copying was prone to G–U errors when creating the positive strand, the same mechanism would be expected to introduce G–U errors when copying the negative strand, resulting in additional C–A errors on the positive strand. Note that in SARS-CoV-2 the G (19.6%) and C (18.4%) content are similar, as are A (29.9%) and U (32.1%) content. Recently it was suggested that SARS-CoV-2 mutation rates could be affected by variations in its RNA-dependent RNA polymerase (Pachetti et al., 2020), but it is unclear how this could explain the asymmetric increase of G–U transversions.

The most known example of mutation bias is the excess of C–T mutations in the CpG context in the genomes of many animals (Cooper & Krawczak, 1989), although other important mutation contexts exist (Panchin et al., 2011). Usually in such cases, the excess of complementary mutations (such as G–A in the case of CpG context) is also present and is of the same magnitude.

In theory, strand-specific RNA editing could cause the observed mutation asymmetry. Recently, Nanopore experiments suggested that SARS-CoV-2 has unique RNA-editing sites (Kim et al., 2020). However, this editing was associated with the second position of the AAGAA motif. We checked if this or any motif was present near G–U mutations in the SARS-CoV-2 genome, but found none (Fig. S2).

Di Giorgio et al. (2020) showed the importance of RNA editing in the SARS-CoV-2 genome-wide mutagenesis. They analyzed multiple cDNA reads of viruses from three patients for signs of RNA editing. They found excessive C–U and G–A SNV’s, which could be derived from human APOBEC-mediated C–U deamination (Blanc & Davidson, 2010). They also found excessive A–G and U–C changes that could be derived from deamination of adenosine to inosine mediated by ADAR (Samuel, 2011). These A–G and U–C changes were the most predominant. However, in one of three patients, with the highest read coverage, G–U SNVs were also abundant. In the other two patients, the read coverage was much lower, so it is difficult to conclude if there was a difference in SARS-CoV-2 mutation rates between patients. Interestingly, the data provided by Di Giorgio et al. (2020) also reveals G–U and C–A mutation asymmetry not only in SARS-CoV-2 but also in MERS-CoV.

One notable cause of G–T mutations in DNA is due to reactive oxygen species that generate 8-oxoguanine (8-oxoG) (Ohno et al., 2014), which can be paired not only with cytosines, but also with adenines, resulting in nucleotide mis-incorporation during DNA synthesis (Dai et al., 2018). The same mechanism may lead to errors during RNA synthesis as well (Li, Wu & Deleo, 2006), perhaps even more so, considering that RNA is more prone to oxidative damage than DNA under similar conditions (Li, Wu & Deleo, 2006). It is known that 8-oxoG is involved in transcriptional mutagenesis (Dai et al., 2018) and oxidative stress is associated with respiratory viral infections (Delgado-Roche & Mesta, 2020). In addition, Schneider et al. (1993) reported 8-oxoG formation in isolated RNA of RNA bacteriophages induced by reactive oxygen species.

If higher levels of 8-oxoG generation were associated with peak concentrations of (+) SARS-CoV-2 RNAs in infected cells (at later stages of the infection at the cellular level), 8-oxoG-rich (+) RNA would transfer to new cells, leading to G–A mispairing during subsequent (−) RNA synthesis. This could hypothetically lead to the observed G–U and C–A mutation asymmetry. Bats have evolved increased resistance to oxidative stress (Chionh et al., 2019), which could explain why the excess of G–U substitutions is not observed between SARS-CoV-2 and bat coronavirus RaTG13. It is unclear, however, why SARS-CoV-2 is different from SARS-CoV in this regard. We believe this hypothesis requires further investigation.

Conclusions

We report a 9-fold asymmetrical G–U (+) strand or C–A (−) strand mutation bias in SARS-CoV-2. This feature cannot be traced in the substitution data that reflects the virus’s evolutionary history before its transmission to our species. The observed effect points to a recently acquired change in SARS-CoV-2 mutation pattern or difference in its pathology in humans and bats. Additional studies are warranted to pinpoint the mechanism by which this mutation bias is introduced and how its asymmetry is maintained.

Supplemental Information

Supplemental Information 1 A list of accessions used in multiple alignments, a simple Neighbor-joining tree for nine different complete coronavirus genomes, coordinates of SARS-CoV-2 G to U transversions and a sequence frequency logo based on nucleotide frequencies surrounding G to.

Click here for additional data file.

Supplemental Information 2 Three multiple alignments are of multiple SARS-CoV-2, SARS-CoV and HKU1 genomes. Three more are multiple alignments of reference human coronaviruses and their relative coronaviruses from bats.

Click here for additional data file.

Supplemental Information 3 Perl scripts.

Two perl scripts used to calculate the number of mutations and substitutions in the multiple alignments.

Click here for additional data file.

We are grateful to Alexander Tyshkovskiy and Evgenia Dueva for helpful comments on our manuscript.

Additional Information and Declarations

Competing Interests

Author Contributions

Data Availability

The authors declare that they have no competing interests.

Alexander Y. Panchin conceived and designed the experiments, performed the experiments, analyzed the data, prepared figures and/or tables, authored or reviewed drafts of the paper, and approved the final draft.

Yuri V. Panchin conceived and designed the experiments, analyzed the data, authored or reviewed drafts of the paper, and approved the final draft.

The following information was supplied regarding data availability:

Raw data and code are available in the Supplemental Files.

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
