# Peer review of "Excessive G–U transversions in novel allele variants in SARS-CoV-2 genomes"

_PeerJ, doi:10.7717/peerj.9648_

## Round 0.1 · original submission · Major Revisions

Your manuscript deals with a very interesting observation that will be of great interest to the scientific community. However, it needs substantial improvement before it can be published. I agree with reviewers #1 and #3 that changes should be expressed as G to U instead as G to T since they are mutations of an RNA virus. The methodology as to how you have obtained the consensus sequence needs to be better explained and reasoned why the ancestral sequence was not used. Also, a statistical analysis must be performed.

Please respond to all the concerns of the reviewers. I believe all their suggestions are very appropriate and will improve your manuscript.

·

Basic reporting

Overall, the manuscript is concise and has an easy to read narrative. A few comments:
Lines 116-121: It would be better to use percentages as well as the raw number ratios, as is done later in the text. Comparing the raw ratios is not very intuitive for the reader.
The phrasing ‘remarkable feature’ in lines 142 and 146 is a bit unclear. Maybe change to ‘observations’?
Line 148: change ‘plus’ to ‘positive’ strand.

The introduction is short and to the point. One misunderstanding of the current literature is that the virus RaTG13 is not technically the closest known relative of SARS-CoV-2 (lines 21-22 & 47-48). That would be the virus RmYN02 (Zhou et al., 2020, Current Biology 30, 1–8; https://doi.org/10.1016/j.cub.2020.05.023). This should be clarified in the introduction.

Mutations and substitutions calculated through both methods used by the authors are presented in tables 1 and 2, and figure 1 clearly showcases the data.

Experimental design

The methodology used by the authors seems reasonable, although some parts are unclear in the manuscript. In lines 66-67 the authors do not explain how they have derived the consensus sequences. They should cite any existing methods they have used, or if inferred with their own code mentioned that they have done so. In the latter case, if the authors believe their method diverges from conventional ways of inferring consensus sequences it might be worth making the code public with this publication.

In lines 82-88 the authors describe the bat virus relatives to the human coronaviruses used in their analysis. I would urge the authors to include a phylogeny of how these viruses are related in their manuscript. That would make the analysis much clearer to audience that is not familiar with the coronavirus phylogeny.

As I point out above RmYN02 is the closest known relative to SARS-CoV-2. The authors might decide to also use this virus in their analysis. In case they do they will have to account for some distinct recombination patterns in RmYN02 described in Zhou et al. (2020). Still, I believe that only using RaTG13 should not make much difference in the findings.

Validity of the findings

The findings seem valid and the increased G to U rate observation in SARS-CoV-2 should prove useful to other researchers exploring similar aspects of the virus. In their discussion the authors mention some rather speculative explanations of their findings (e.g. lines 178-181), nonetheless the phrasing makes it clear that these are speculations and the authors rightfully point out that further research on the topic needs to be conducted to support such hypotheses.

Additional comments

In this paper the authors quantify and compare the single nucleotide mutation rates and substitution rates between different members of the Coronaviridae family, focusing on SARS-CoV-2, a pathogen that is currently of paramount interest. The authors highlight the interesting observation that G to U changes tend to be more frequent in SARS-CoV-2 after its emergence in humans compared to its bat relative and other coronaviruses that have previously crossed to humans.
I believe that this observations could be of interest to the scientific community working on similar aspects of the SARS-CoV-2 virus.

Reviewer 2 ·

Basic reporting

- References are provided.
- Relevant figures and tables are provided.

Experimental design

- Research questions are will defined.
- Methods are appropriately described.

Validity of the findings

- Conclusions are appropriately stated.

Additional comments

The manuscript of Panchin A and Panchin Y "Excessive G to T transversions in novel allele variants in SARS-CoV-2 genomes" is devoted to the analysis of the mutation and substitution spectrum in new SARC-CoV-2 coronavirus. Unprecedented attention to this virus causing the serious infection diseases COVID-19 leads to a large number of the sequenced genomes allowing to track the evolution of the virus in human. Authors investigated the mutations patterns of the SARS-CoV-2 and found the elevated number of the G to T mutations possibly reflecting the adaptation of the virus in human. Taking into account the urgent need for rapid publications of the studies of the SARS-CoV-2 the quality of the manuscript in general satisfactory, although I would suggest some improvements.
To clarify the nature of the G>T transversions I would suggest to answer in the manuscript the following questions:
1. What is the number of G>T transversions are located in coding and non-coding regions?
2. Among the G>T transversions in coding regions what is the number of synonymous and non-synonymous mutations?
3. Among the non-synonymous G>T transversions what is the number of missense and nonsense mutations.
4. What is the nucleotide context (5' and 3' nucleotide) of the observed G>T transversions?
5. The multiple alignment should be attached as the supplemental file.
6. Authors should also support the manuscript with the supplemental file containing the list of the observed G>T transversions and its genome positions.

Reviewer 3 ·

Basic reporting

Panchin & Panchin present a comparison of nucleotide variation patterns in SARS-CoV-2 publicly available genomes. They key finding is a 9-fold excess of G to T (I personally prefer U) transversion. Not a lot of discussion is given about the mechanism for this excess, only speculating a role of the reactive oxygen species mutagenic base 8-oxoguanine (8-oxoG).

Please revise the literature for preprint already published. Like in the case of ref.18: https://advances.sciencemag.org/content/early/2020/05/15/sciadv.abb5813

I want to encourage the authors to use U instead of T across this manusctipt: title, text, tables and figure.

Experimental design

Overall, is a consice work that could be expanded. One major comment is whether the methodology is adequate. I believe that determining the ancestral state as the consensus sequence lacks biological correction. Why don’t the authors use the date of the samples to estimate these ancestral states? Sequences from december 2019-early january 2020 will certainly be a better proxy to estimate these N1 states (N1s) than the consensus. Tables 1 and 2 remain similar when looking at only at the first sequences availabable to determine N1s?

Another thing to take into account is how N2 changes are calculated. Every time in every genome that have a base different to N1 is counted? If thats is the case, it is likely that the authors are counting the same change many times. If that is the case, this big excess in G to U could be an artefact. An elegant way to adress this issue is to build a philogeny and count the changes that occur in wranches throughout the tree. This is a much better way of represent what is happening in reality. These viruses are from different time points and this temporality must be taken into account.

If this patterns are robust after the considerations given above, the next big step is to show where in the genome those mutations occur. They occur across the whole genome or are prevalent in some regions?

Validity of the findings

Statistical analyses are lacking. I suggest Chi squared analysis or Fisher’s exact test between groups to see if there are any significant differences, but another test could also be used.

About the 8-oxoG: A more extensive review of the molecular mechanism of this base modification and the machinery involved is very wellcome. More details about how and in what degree this mechanism can affect viral transcription and viral replication.

Additional comments

Another comment is that the data and code used in this analysis must be presented. A list of accessions is missing and the code is a good way of check the preocedure, besides the one described in Methods section. The alignments also must be presented. Given the big size of this genomes, I suggest the use more suitable algorithms than Clustal Omega (i.e.; MAFFT, BAli-Phy).

In lines 103 and 107-108, the authors mention reference genomes. Please specify the accession number of these reference genomes. Is it the RefSeq genome or the more similar to N1s? Again, if a list of accession is given (along with the code), these methodological questions will be more easy to be responded directly.

---

## Round 0.2 · accepted · Accept

Your manuscript has been adequately corrected following the reviewers' comments and suggestions. In the new version the experimental design and methodology are well described and your findings are experimentally and statistically sound. I believe your work is now ready for publication.

·

Basic reporting

The authors have addressed all changes requested, clarifying the relation between SARS-CoV-2, RaTG13 and RmYN02 and revising few minor phrasing and reporting suggestions.

Experimental design

In this version of the submission the authors have adequately described the experimental design and methodology and provided extra supplementary material essential for transparency and reproducibility of their research.

Validity of the findings

As mentioned in the first round of review the findings are interesting and valid in the way they are presented and discussed.

Additional comments

I believe that the authors have addressed all comments successfully and the manuscript now looks very presentable.

Reviewer 3 ·

Basic reporting

I would like to highlight that all reviewers' comments have been properly addressed. I would also like to mention that there was considerable harmony in these comments, which clearly facilitates this process.

Experimental design

Again, I think the comments have been answered correctly.

As an extra comment (no response is needed), I want to add that the 193 positions reported on Table S2 also correspond to a G on the SARS-CoV2 reference sequence (NC_045512).

Validity of the findings

The G to U pattern observed in this virus is interesting. In addition to its magnitude, the authors also present statistics.

The authors add a few more references on 8-oxoguanine. I agree that it is a topic that is not very well known, let alone related to viruses.

Additional comments

This version, although it has small changes, is sufficiently improved. It is also better supported by supplementary material.